# Tree of Attributes Prompt Learning for Vision-Language Models

## Abstract

Prompt learning has proven effective in adapting vision language models for downstream tasks. However, existing methods usually append learnable prompt tokens solely with the category names to obtain textual features, which fails to fully leverage the rich context indicated in the textual category name. To address this issue, we propose the Tree of Attributes Prompt learning (TAP), which first instructs LLMs to generate a tree of attributes with a "concept - attribute - description" structure for each associated category name, and then learn the hierarchy with vision and text prompt tokens. Unlike existing methods that merely augment category names with a set of unstructured descriptions, our approach essentially distills structured knowledge graphs associated with class names from LLMs. Furthermore, our approach introduces text and vision prompts designed to explicitly learn the corresponding visual attributes, effectively serving as domain experts. Additionally, the general and diverse descriptions generated based on the class names may be wrong or absent in the specific given images. To address this misalignment, we further introduce a vision-conditional pooling module to extract instance-specific text features. Extensive experimental results demonstrate that our approach outperforms state-of-the-art methods on the zero-shot base-to-novel generalization as well as few-shot classification across 11 diverse datasets.

## 1 Introduction

Recent advancements in vision-language models (VLMs) like CLIP [33] and ALIGN [13] merge the capabilities of visual perception with linguistic understanding, which have revolutionized the landscape with their zero-shot learning abilities. They proficiently handle tasks on unseen data, bypassing the conventional requirement for task-specific training. This feature has enabled a plethora of applications, ranging from content-based image retrieval to complex visual question answering, setting new benchmarks in the domain. A crucial development in this domain is the concept of prompt learning, which has significantly influenced both natural language processing (NLP) [20–22] and vision-only models [14, 43, 44, 51]. This approach leverages learnable prompts to guide model understanding, tailoring responses to specific tasks or datasets.

Prompt learning, particularly in vision-language models, has garnered considerable interest due to its parameter efficiency and rapid convergence [54, 53, 55, 8, 23]. Techniques like CoOp [54] optimize learnable continuous prompts for few-shot image recognition, enhancing model performance significantly. Recent efforts have expanded to multimodal prompt learning, optimizing prompts in both visual and language domains [15, 16, 38, 19]. Despite their success, these models rely on simplistic text prompts, typically formatted as "a photo of a {class}", illustrated in Fig. 1 (a). While functional, this approach lacks depth, failing to encapsulate the intricacies and finer details inherent in

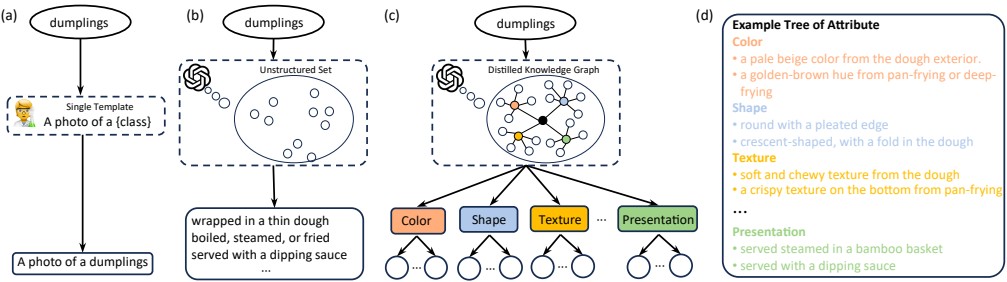

Figure 1: Illustration of the methods for CLIP text prompts formation. (a) Manually created prompt with the single "a photo of a {class}" template; (b) A unstructured set of detailed descriptions generated by LLMs; (c) The proposed Tree of Attribute that organizes the descriptions in a "concept - attribute - descriptions" structure, essentially distilling knowledge graphs from LLMs; (d) An example Tree of Attribute for "dumplings".

visual data. Such limitations hinder the model's ability to fully leverage the rich, descriptive potential offered by more detailed and contextually relevant textual information.

In parallel, another stream of research has been exploring the utilization of large language models (LLMs) to generate more elaborate and descriptive text prompts for enhancing zero-shot learning capabilities [26, 32, 35, 17, 30, 48, 49, 36, 52, 40]. These LLM-generated descriptions offer a wealth of detail and context, potentially enriching the model's interpretative capabilities. However, current methodologies in integrating these descriptions often do not exploit the full potential of this richness. As shown in Fig. 1 (b), most of these approaches lack a structured framework to organize and utilize these descriptions effectively, leading to a scattergun approach where not all generated descriptions are contextually relevant or optimally aligned with the visual content. In addition, as noted in [35], descriptions generated by such paradigms are usually diverse, which covers most possibilities of the class, but include descriptions that are either likely not co-occurring, e.g. "steamed" and "fried", or absent in the input image, e.g. "long tail" for a cat shot from the front, necessitating the need for a selective pooling mechanism for clearer image-text alignments.

In response to these challenges, our work introduces "Tree of Attribute Prompt learning (TAP)," a method that redefines the integration and utilization of detailed descriptions within VLMs. As indicated in Fig. 1 (c), unlike existing methods that merely augment category names with a set of unstructured descriptions, our approach essentially distills structured knowledge graphs associated with class names from LLMs. Specifically, we adopt a hierarchical, tree-like structure to systematically generate and integrate descriptions, ensuring a layered and comprehensive understanding of visual content. Each branch of this tree represents a specific attribute, with finer details fleshed out in the subsequent leaves, ensuring that every aspect of the visual content is captured and represented. Furthermore, we reimagine the learnable prompt tokens as "domain experts", each specializing in different aspects of the image, supplemented by the CLS token's global perspective. In addition, we introduce vision-conditional layers for each expert-attribute pair, which pool the most applicable descriptions from each of the attribute sets with condition on the input image content, ensuring optimal image-text alignment. This setup not only provides a detailed, attribute-focused analysis but also harmonizes these insights with the overall context.

Extensive experiments in both base-to-novel generalization and few-shot classification across 11 diverse datasets demonstrate the effectiveness of our method. On base-to-novel generalization, TAP achieves average performance gains of $1.07\%$ in harmonic mean over the state-of-the-art methods, and $9.34\%$ over the vanilla CLIP. Competitive results are also observed in few-shot classification.

## 2   Related Work

**Prompt Learning for Vision-Language Models.** Prompt learning bridges linguistic understanding and visual perception by guiding VLMs with text prompts, a concept originated in NLP [20–22] and adapted to vision-only [14, 43, 44, 51] and multimodal contexts[54, 53, 15, 16, 38, 19, 40, 34, 36, 52, 55, 4, 23]. In the textual domain, CoOp [54] optimizes learnable continuous prompts in CLIP's language branch for few-shot image recognition, while CoCoOp [53] addresses CoOp's

overfitting issues by conditioning prompts on visual features. In the visual domain, Visual Prompt Tuning (VPT) [1] and Dual-modality Prompt Tuning (DPT) [47] enhance CLIP's vision encoder by learning visual prompts in pixel space and dynamically generating prompts through cross-attention, respectively. TransHP [42] leverages category hierarchy for prompt learning to improve classification performance. LoGoPrompt [38] enhances classification by incorporating synthetic images with class name text as auxiliary visual prompts. MaPLe [15] explores multimodal prompt learning, jointly optimizing prompts in both vision and language branches. Other recent works have focused on regularizing prompt learning to leverage the knowledge from base VLMs effectively, demonstrating enhanced generalization in varied downstream visual tasks [16, 4, 36]. PromptSRC, for instance, introduced a self-regulating method that restricts both the vision and text prompt, demonstrating improved generalization. Distinct from these approaches, PLOT [5] and ALIGN [41] leverage Optimal Transport to align multiple prompts with local visual features, either from the multi-head self-attention layer or at a token level. Our work diverges from these methods by introducing a hierarchical "Tree of Attribute" framework derived from LLMs to structure textual descriptions and guide the learning of specialized "domain expert" tokens for attribute-level understanding.

**Image classification by descriptions.** There's a growing emphasis on using visual descriptions for zero-shot recognition, moving beyond generic prompts [54, 53]. These descriptions, like the "fur pattern" or "tail shape" of a cat, provide fine-grained and distinctive characteristics. The use of LLMs like GPT-3 [3], allows for efficient generation of a broad spectrum of class-specific descriptions, offering an advantage over manually crafted templates. While this approach has been extensively researched in zero-shot contexts [17, 26, 30, 35, 48, 49, 10, 32, 28], its application in conjunction with prompt learning for few-shot tasks remains relatively unexplored[25, 19, 40, 52, 50]. Previous methodologies, however, have largely utilized unstructured descriptions, lacking an organized framework for effective utilization. Our approach diverges by structuring these descriptions into a "Tree of Attribute" model, coupled with learnable visual prompts as domain experts. Additionally, LLM-generated descriptions often cover a wide range of potential class descriptions, of which not all may be pertinent to a given image, pointing to the need for a selective pooling mechanism to ensure optimal image-text alignment. We further introduce a vision-conditional pooling layer for refined image-text alignment. This structured approach not only enhances the interpretability of the model's learning process but also significantly improves alignment accuracy between image content and descriptive text.

## 3 Methodology

### 3.1 Preliminary

**CLIP.** Our approach is built on the pre-trained vision-language model, CLIP [33]. Formally, let $(x, c)$ denote the dataset, where $x$ is an image and $c \in \{1, \ldots, C\}$ are the class labels. For an image $x$, the vision encoder $h_I(\cdot)$ transforms it into a feature vector $\mathbf{f}_x^v = h_I(x)$. Simultaneously, each class label $c$ is mapped to a text prompt $t_c = $ a photo of a {c}, and converted into textual feature vectors $\mathbf{f}_c^t = h_T(t_c)$. The predicted class $\hat{y}$ is given by:

$$\hat{y} = \underset{c}{\operatorname{argmax}} \cos(\mathbf{f}_x^v, \mathbf{f}_c^t) \tag{1}$$

where $\cos(\cdot)$ denotes cosine similarity.

**Image classification with class descriptions.** To improve the model's understanding of the categories in the transfer datasets, previous works [26, 35] use more detailed descriptions from Large Language Models (LLMs) instead of the simple "a photo of a {c}" to prompt the CLIP text encoder. Under this approach, a convoluted set of descriptions is generated for a class $c$ as $\mathcal{D}_c : \{$"c, which is/has/etc description." $\}$, e.g. c="television" and description="black or grey". This classification is reformulated as

$$\hat{y} = \underset{c}{\operatorname{argmax}} \frac{1}{|\mathcal{D}_c|} \sum_{d \in \mathcal{D}_c} \cos(\mathbf{h_I}(x), \mathbf{h_T}(d)) \tag{2}$$

### 3.2 Overall Framework

We rethink the descriptions by LLM $\mathcal{D}_c$ as nodes in knowledge graphs. While previous methods generate an unstructured set of descriptions, we distill structured knowledge graphs for each class $c$

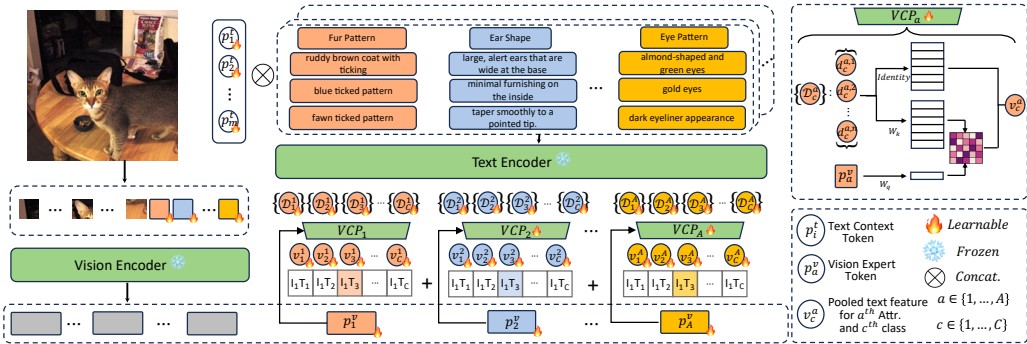

Figure 2: Overview of the proposed TAP method. TAP utilizes fine-grained descriptions from LLMs and organizes them in a Tree of Attribute. Vision expert tokens are added to the vision encoder to learn from specific attributes such as color and shape. A vision-conditional pooling layer is introduced to ensure optimal image-text alignment. Textual context tokens are also incorporated to the textual branch, shared across descriptions.

from LLM, in which the root node is the class name $c$, capturing the highest level semantics, and the leaf nodes are the detailed descriptions capturing fine-grained details. In this framework, previous paradigms only generate the leaf nodes of the graph, with the edges and graph structure missing, where the rich and inherent structure from the descriptions is overlooked. To address this limitation, we formulate our approach as a Tree of Attribute, which follows the "concept - attribute - description" structures, as illustrated in Fig. 1 (c).

Besides weighting the descriptions equally, previous works typically align descriptions that describe images from different aspects and at different granularities with a singular CLS token from the image encoder. However, while the use of a single CLS token is effective in certain contexts, we note that the CLS token is designed to capture the global information of an input image $x$ [9]. As a result, even though this helps to further inform global understanding, it may fail to effectively capture the nuances and variances at the attribute level. This leads to suboptimal use of the rich descriptions. We address this by introducing a set of learnable prompt tokens that serve as domain experts in the vision branch, each of which aligns with a specific attribute-level textual embedding.

Additionally, close inspection of the LLM-generated descriptions indicates limited contextual relevance and a high degree of diversity. Previous works [35] reflect the issue of descriptions that are likely not co-occurring e.g. "steam" and "fried". We further identify cases where the descriptions are technically correct but irrelevant to certain images, such as describing "long tail" in frontal images of cats, underscoring the need for a selective pooling mechanism. Thus, we introduce a vision-conditional pooling layer to extract instance-specific text features for each attribute for selecting the most applicable descriptions.

Overall, our approach utilizes fine-grained descriptions and organizes them in a Tree of Attribute following the "concept - attributes -descriptions" structure. Learnable vision expert tokens are appended to the input image embedding to learn from specific fine-grained attributes such as color and shape. A vision-conditional pooling layer is further added for each attribute to ensure optimal image-text alignment. Inspired by CoOP [54], we also incorporate textual contextual tokens in the text encoder. The overall framework is presented in Fig. 2.

### 3.3 Tree of Attribute generation by LLMs

We redefine the process of integrating LLM-generated descriptions by introducing a knowledge graph $\mathcal{G}_c = \{\mathcal{V}_c, \mathcal{E}_c\}$ for each class $c$, where $\mathcal{V}_c$ denotes the set of nodes, and $\mathcal{E}_c$ denotes the edges that capture the semantic relationship between nodes. In previous works, $\mathcal{V}_c$ is the set of descriptions $\mathcal{D}_c$, while $\mathcal{E}_c$ is missing. We argue that such methods overlook the inherent structure among the descriptions and thus do not exploit the richness of these descriptions effectively. To better leverage knowledge from LLMs, we introduce an attribute layer to link the root node class name, and the leaf node descriptions. The attribute nodes include visual attributes generated by LLMs, such as color and shape, for systematically guiding description generation as illustrated in Fig. 1 (c). Each branch of this "tree" represents a specific attribute, with the subsequent "leaves" fleshing out the descriptions

with finer details. In this framework, $\mathcal{V}_c$ includes the class name which is the root node, the set of attributes such as color and shape being the intermediate layer, and lastly the set of descriptions under each attribute node. $\mathcal{E}_c$ includes the edges that build up the hierarchy. This structure allows for a nuanced representation of class information, spanning from general concepts down to specific attributes and detailed descriptions.

To this end, we introduce the Tree of Attribute (ToA), where we use a tree structure to model the relationship and structure of the descriptions. Let $\mathcal{A}_c$ denote the set of attributes, and for each attribute $a_c \in \mathcal{A}_c$, we denote its leaf nodes as $\mathcal{D}_c^a$. Each set $\mathcal{D}_c^a$ contains descriptions that specifically pertain to attribute $a$ for class $c$, which is denoted as

$$\mathcal{D}_c^a = \{d_c^{a,1}, d_c^{a,2}, \ldots, d_c^{a,n}\}, \tag{3}$$

where $d_c^{a,i}$ represents the $i$-th description for attribute $a$ of class $c$ and $n$ is the number of descriptions per attribute.

The process of generating a Tree of Attribute (ToA) unfolds in three steps: 1) **Attribute Generation:** We first query LLMs with the dataset information and ask it to generate a set of attributes $\mathcal{A}$ which are considered relevant and characteristic of the dataset. 2) **Example Generation:** We then ask LLMs to generate descriptions for a randomly sampled class in the dataset, using the attributes $\mathcal{A}$ identified in the previous step. Each description takes the format of "class, which {is/has/etc} {description}". Human review is performed to ensure the quality of the example. 3) **Description Generation for All Classes:** Building upon the Q&A template from the previous step, the LLM is then tasked with generating descriptions for all classes in the dataset.

Additionally, we incorporate a "global context" attribute which is aligned with the CLS token in the vision encoder. The descriptions are the 7 standard templates provided in [33].

## 3.4 Learning TAP with Learnable Expert Tokens

To fully exploit the structured Tree of Attribute, we introduce learnable visual expert tokens $\mathbf{p}_a^v$ in the vision branch to learn from each of the attribute nodes $a \in \mathcal{A}$. Unlike traditional methods that rely on a single CLS token for alignment, these expert tokens enable focused learning on specific image attributes, such as color or shape, enhancing the model's performance and interpretability.

We denote the set of introduced visual expert tokens as $\mathcal{P}^v = \{\mathbf{p}_a^v | a \in \mathcal{A}\}$. Akin to the idea of visual prompt tuning (VPT) [14], we insert $\mathcal{P}^v$ into the input sequence of the vision encoder, forming the prompted input sequences $\tilde{\mathbf{X}}_{\mathbf{P}} = \{\mathbf{e}_{\text{CLS}}, \mathcal{P}^v, \mathbf{E}_{\text{patch}}\}$, where $\mathbf{e}_{\text{CLS}}$ is the input CLS token, and $\mathbf{E}_{\text{patch}}$ denotes the embedded patch tokens. To further boost the model's capacity for nuanced attribute representation, we employ deep prompting by introducing a zero-initialized layer residual for each prompt token across transformer layers, which provides more explicit attribute guidance across transformer layers. In parallel, we adopt a set of $m$ learnable context tokens $\mathcal{P}^t = \{\mathbf{p}_j^t | j \in \{1, 2, ..., m\}\}$ for the text encoder shared across all descriptions, similar to [54].

## 3.5 Vision-Conditional Pooling

To mitigate issues of misalignment and potential misleading information from the broad spectrum of LLM-generated descriptions, we proposed an adaptive vision-conditional pooling layer, applicable to each set of attribute descriptions $\mathcal{D}_a$ shared across all classes to dynamically pool the most applicable descriptions based on the visual content of the image $x$ using its corresponding visual expert token denoted as $\mathbf{p}_{a,x}^v$. For ease of expression, we will proceed without explicitly mentioning $x$, though it's important to note that both the expert token and the resulting attribute-level embeddings are dependent on the visual information. Intuitively, VCP uses attention to calculate the similarity between $\mathbf{p}_a^v$ and all embedded descriptions in attribute $\mathcal{D}_a$, which are then used as weights for a weighted sum of the original description embeddings. Formally, for each attribute $a$ and its associated expert token $\mathbf{p}_a^v$, the pooled attribute-level embedding $\mathbf{v}_c^a$ for class $c$ and attribute $a$ is:

$$\begin{aligned}
\text{Query} &= W_q \cdot \mathbf{p}_a^v, \\
\text{Key} &= W_k \cdot \text{Emb}(\mathcal{D}_c^a), \\
\text{Attention Score} &= \text{softmax}(\text{Query} \cdot \text{Key}^T), \\
\mathbf{v}_c^a &= \text{Attention Score} \cdot \text{Emb}(D_c^a),
\end{aligned} \tag{4}$$

where $W_q$ and $W_k$ are learnable weights $\in \mathbb{R}^{d \times d}$, $\texttt{Emb}(\cdot)$ denotes the embedding function, and $\texttt{softmax}(\cdot)$ is the Softmax function. This layer mirrors cross-attention but omits $W_v$ to maintain the output within the CLIP V-L space.

## 3.6 Training and Inference

**Training objective.** During training, each visual expert token $\mathbf{p}_a^v$ is aligned with its associated attribute-level embedding $\mathbf{v}_c^a$, trained with the following contrastive objective:

$$L_{con}(\mathbf{p}_a^v, \mathbf{v}_c^a) = -\frac{1}{N} \sum_{i=1}^{N} \log \frac{\exp(\cos(\mathbf{p}_a^v, \mathbf{v}_y^a)/\tau)}{\sum_{c=1}^{C} \exp(\cos(\mathbf{p}_a^v, \mathbf{v}_c^a)/\tau)}, \tag{5}$$

where $N$ represents the number of training samples, and $\tau$ is the learned temprature of CLIP. The total classification loss $L_{\text{class}}$ is the average of the contrastive loss from each expert token as well as the CLS token, defined as:

$$L_{class} = \frac{1}{|\mathcal{A}|} \left( \sum_{a \in \mathcal{A}} L_{con}(\mathbf{p}_a^v, \mathbf{v}_c^a)) \right), \tag{6}$$

Similar to [16] and [4], we regularize the vision CLS token, text feature, and the prediction logits from each attribute using the vanilla CLIP model. We denote the regularization loss as $L_{reg}$, where the details can be found in Appendix. The overall training objective is $L_{\text{total}} = L_{\text{class}} + L_{\text{reg}}$.

**Prediction fusion.** During inference, we integrate the prediction by each attribute expert pair by a weighted sum, formulated as follows:

$$\tilde{y} = \underset{c}{\arg\max} \left( \alpha \cos(\mathbf{f}_{CLS}^v, \mathbf{v}_c^{CLS}) + \frac{1-\alpha}{|\mathcal{A}| - 1} \sum_{a \in \mathcal{A} \backslash \{CLS\}} \cos(\mathbf{p}_a^v, \mathbf{v}_c^a) \right) \tag{7}$$

where $\alpha$ is a hyperparameter that signifies the weight assigned to the global context provided by the CLS token, balancing its contribution with that of the attribute-specific expert prompts.

## 4 Experiments

We extensively evaluate our method in two settings: 1) Base-to-novel class generalization, where the datasets are equally split into base and novel classes. We train the model on the base classes only and evaluate on both base and novel classes; and 2) Few-shot classification with 16 shots per class.

**Datasets and baslines.** For both base to novel class generalization and few-shot setting, we follow previous works [54, 53], using 11 image recognition datasets. The datasets span a range of recognition tasks: ImageNet [7] and Caltech101 [11] for generic object recognition; OxfordPets [30], StanfordCars [18], Flowers102 [27], Food101 [2], and FGVCAircraft [24] for fine-grained classification; SUN397 [46] for scene recognition; UCF101 [39] for action recognition; DTD [6] for texture classification; and EuroSAT [12] for satellite image analysis. We benchmark against several leading methods, including CLIP [33], CoOp [54], Co-CoOP [53], ProGrad [55], RPO [19], LoGoPrompt [38], and the state-of-the-art PromptSRC [16].

**Implementation details.** A pre-trained CLIP model with a ViT-B/16 vision backbone is used in all of our experiments and results are averaged over 3 runs. We use GPT-3.5-turbo [29] for attribute and description generation. We initialize the text context tokens with the word embedding of $\texttt{a photo}$ $\texttt{of a.}$ For both settings, we iteratively train the vision and text encoders with 5 epochs for vision and 1 epoch for text schedule. We set $\alpha = 0.4$, $\mu_1 = 10$, and $\mu_2 = 2.5$ for all datasets. We train the vision encoder for 50 and 100 epochs, and text encoder for 10 and 20 epochs for base-to-novel generalization and few-shot experiments, respectively. For DTD, Oxford Flowers, Stanford Cars, UCF101, and Caltech101 datasets, we use a learning rate of 0.002 for the text encoder and 0.006 for the vision encoder, with $\mu_3 = 3$. For the remaining 6 datasets, the learning rates for both text and vision encoders are set as 0.004, with $\mu_3 = 1.5$. We also use a Gaussian Prompt Weighting (GPA) following [16], with a mean of 45, std of 10 for base-to-novel generalization, and 80, 20 for few-shot experiments. Refer to the Appendix for additional implementation details.

Table 1: **Comparison of TAP in base-to-novel generalization**. HM: harmonic mean [45].

**(a) Average**

| | Base | Novel | HM |
|---|---|---|---|
| CLIP | 69.34 | 74.22 | 71.70 |
| CoOp | 82.69 | 63.22 | 71.66 |
| Co-CoOp | 80.47 | 71.69 | 75.83 |
| ProGrad | 82.48 | 70.75 | 76.16 |
| RPO | 81.13 | 75.00 | 77.78 |
| LoGoPrompt | 84.47 | 74.24 | 79.03 |
| PromptSRC | 84.26 | 76.10 | 79.97 |
| TAP | **84.75** | **77.63** | **81.04** |

**(b) ImageNet**

| | Base | Novel | HM |
|---|---|---|---|
| CLIP | 72.43 | 68.14 | 70.22 |
| CoOp | 76.47 | 67.88 | 71.92 |
| Co-CoOp | 75.98 | 70.43 | 73.10 |
| ProGrad | 77.02 | 66.66 | 71.46 |
| RPO | 76.60 | 71.57 | 74.00 |
| LoGoPrompt | 76.74 | **70.83** | 73.66 |
| PromptSRC | 77.60 | 70.73 | **74.01** |
| TAP | **77.97** | 70.40 | 73.99 |

**(c) Caltech101**

| | Base | Novel | HM |
|---|---|---|---|
| CLIP | 96.84 | 94.00 | 95.40 |
| CoOp | 98.00 | 89.81 | 93.73 |
| Co-CoOp | 97.96 | 93.81 | 95.84 |
| ProGrad | 98.02 | 93.89 | 95.91 |
| RPO | 97.97 | 94.37 | 96.03 |
| LoGoPrompt | 98.19 | 93.78 | 95.93 |
| PromptSRC | 98.10 | 94.03 | 96.02 |
| TAP | **98.90** | **95.50** | **97.17** |

**(d) OxfordPets**

| | Base | Novel | HM |
|---|---|---|---|
| CLIP | 91.17 | 97.26 | 94.12 |
| CoOp | 93.67 | 95.29 | 94.47 |
| Co-CoOp | 95.20 | 97.69 | 96.43 |
| ProGrad | 95.07 | 97.63 | 96.33 |
| RPO | 94.63 | 97.50 | 96.05 |
| LoGoPrompt | **96.07** | 96.31 | 96.18 |
| PromptSRC | 95.33 | 97.30 | 96.30 |
| TAP | 95.80 | **97.73** | **96.76** |

**(e) StanfordCars**

| | Base | Novel | HM |
|---|---|---|---|
| CLIP | 63.37 | 74.89 | 68.65 |
| CoOp | 78.12 | 60.40 | 68.13 |
| Co-CoOp | 70.49 | 73.59 | 72.01 |
| ProGrad | 77.68 | 68.63 | 72.88 |
| RPO | 73.87 | **75.53** | 74.69 |
| LoGoPrompt | 78.36 | 72.39 | 75.26 |
| PromptSRC | 78.27 | 74.97 | 76.58 |
| TAP | **80.70** | 74.27 | **77.35** |

**(f) Flowers102**

| | Base | Novel | HM |
|---|---|---|---|
| CLIP | 72.08 | **77.80** | 74.83 |
| CoOp | 97.60 | 59.67 | 74.06 |
| Co-CoOp | 94.87 | 71.75 | 81.71 |
| ProGrad | 95.54 | 71.87 | 82.03 |
| RPO | 94.13 | 76.67 | 84.50 |
| LoGoPrompt | **99.05** | 76.52 | **86.34** |
| PromptSRC | 98.07 | 76.50 | 85.95 |
| TAP | 97.90 | 75.57 | 85.30 |

**(g) Food101**

| | Base | Novel | HM |
|---|---|---|---|
| CLIP | 90.10 | 91.22 | 90.66 |
| CoOp | 88.33 | 82.26 | 85.19 |
| Co-CoOp | 90.70 | 91.29 | 90.99 |
| ProGrad | 90.37 | 89.59 | 89.98 |
| RPO | 90.33 | 90.83 | 90.58 |
| LoGoPrompt | 90.82 | 91.41 | 91.11 |
| PromptSRC | 90.67 | 91.53 | 91.10 |
| TAP | **90.97** | **91.83** | **91.40** |

**(h) FGVCAircraft**

| | Base | Novel | HM |
|---|---|---|---|
| CLIP | 27.19 | 36.29 | 31.09 |
| CoOp | 40.44 | 22.30 | 28.75 |
| Co-CoOp | 33.41 | 23.71 | 27.74 |
| ProGrad | 40.54 | 27.57 | 32.82 |
| RPO | 37.33 | 34.20 | 35.70 |
| LoGoPrompt | **45.98** | 34.67 | 39.53 |
| PromptSRC | 42.73 | **37.87** | **40.15** |
| TAP | 44.40 | 36.50 | 40.06 |

**(i) SUN397**

| | Base | Novel | HM |
|---|---|---|---|
| CLIP | 69.36 | 75.35 | 72.23 |
| CoOp | 80.60 | 65.89 | 72.51 |
| Co-CoOp | 79.74 | 76.86 | 78.27 |
| ProGrad | 81.26 | 74.17 | 77.55 |
| RPO | 80.60 | 77.80 | 79.18 |
| LoGoPrompt | 81.20 | 78.12 | 79.63 |
| PromptSRC | 82.67 | 78.47 | 80.52 |
| TAP | **82.87** | **79.53** | **81.17** |

**(j) DTD**

| | Base | Novel | HM |
|---|---|---|---|
| CLIP | 53.24 | 59.90 | 56.37 |
| CoOp | 79.44 | 41.18 | 54.24 |
| Co-CoOp | 77.01 | 56.00 | 64.85 |
| ProGrad | 77.35 | 52.35 | 62.45 |
| RPO | 76.70 | 62.13 | 68.61 |
| LoGoPrompt | 82.87 | 60.14 | 69.70 |
| PromptSRC | 83.37 | 62.97 | 71.75 |
| TAP | **84.20** | **68.00** | **75.24** |

**(k) EuroSAT**

| | Base | Novel | HM |
|---|---|---|---|
| CLIP | 56.48 | 64.05 | 60.03 |
| CoOp | 92.19 | 54.74 | 68.69 |
| Co-CoOp | 87.49 | 60.04 | 71.21 |
| ProGrad | 90.11 | 60.89 | 72.67 |
| RPO | 86.63 | 68.97 | 76.79 |
| LoGoPrompt | **93.67** | 69.44 | 79.75 |
| PromptSRC | 92.90 | 73.90 | 82.32 |
| TAP | 90.70 | **82.17** | **86.22** |

**(l) UCF101**

| | Base | Novel | HM |
|---|---|---|---|
| CLIP | 70.53 | 77.50 | 73.85 |
| CoOp | 84.69 | 56.05 | 67.46 |
| Co-CoOp | 82.33 | 73.45 | 77.64 |
| ProGrad | 84.33 | 74.94 | 79.35 |
| RPO | 83.67 | 75.43 | 79.34 |
| LoGoPrompt | 86.19 | 73.07 | 79.09 |
| PromptSRC | 87.10 | 78.80 | 82.74 |
| TAP | **87.90** | **82.43** | **85.08** |

Table 2: Few shot classification results with 16 shots.

| | Average | ImageNet | Caltech101 | Pets | Cars | Flowers | Food101 | Aircraft | SUN397 | DTD | EuroSAT | UCF101 |
|---|---|---|---|---|---|---|---|---|---|---|---|---|
| | 16-Shot Classification | | | | | | | | | | | |
| CLIP | 78.79 | 67.31 | 95.43 | 85.34 | 80.44 | 97.37 | 82.90 | 45.36 | 73.28 | 69.96 | 87.21 | 82.11 |
| CoOp | 79.89 | 71.87 | 95.57 | 91.87 | 83.07 | 97.07 | 84.20 | 43.40 | 74.67 | 69.87 | 84.93 | 82.23 |
| CoCoOp | 74.90 | 70.83 | 95.16 | 93.34 | 71.57 | 87.84 | 87.25 | 31.21 | 72.15 | 63.04 | 73.32 | 78.14 |
| MaPLe | 81.79 | 72.33 | 96.00 | 92.83 | 83.57 | 97.00 | 85.33 | 48.40 | 75.53 | 71.33 | 92.33 | 85.03 |
| PSRC | 82.87 | 73.17 | 96.07 | 93.67 | 83.83 | 97.60 | 87.50 | **50.83** | 77.23 | 72.73 | **92.43** | 86.47 |
| TAP | **83.37** | **73.76** | **96.73** | **93.90** | **85.37** | **98.10** | **87.53** | 50.43 | **77.30** | **74.90** | 91.90 | **87.17** |

## 4.1 Base-to-Novel Generalization

In base-to-novel generalization, we equally split the classes into base and novel classes. Initial training and evaluations are conducted on the seen base classes, followed by evaluation on the unseen novel classes in a zero-shot manner. TAP surpasses prior state-of-the-art models in terms of the base and novel class accuracy, as well as their harmonic mean across most of the 11 datasets, with an average increase of 1.53% in the zero-shot novel class prediction, and a 1.07% increase in the overall harmonic mean in average, as detailed in Table 1. Notably, our method improves unseen class prediction without compromising base class performance, exhibiting an average performance boost of 0.49%. In the challenging fine-grained tasks such as DTD, EuroSAT, and UCF101, TAP achieves significant improvements in novel class prediction by 5.03%, 8.27%, and 3.63% respectively. These results underscore the robust generalizability and efficacy of our method across diverse scenarios.

## 4.2 Few-Shot Classification

In few-shot classification, TAP also outperforms existing methods in 9 out of the 11 datasets. Detailed in Table 2, we achieve an average accuracy of 83.37 across the 11 datasets, surpassing the previous state-of-the-art methods by 0.5%, further demonstrating the effectiveness of our method.

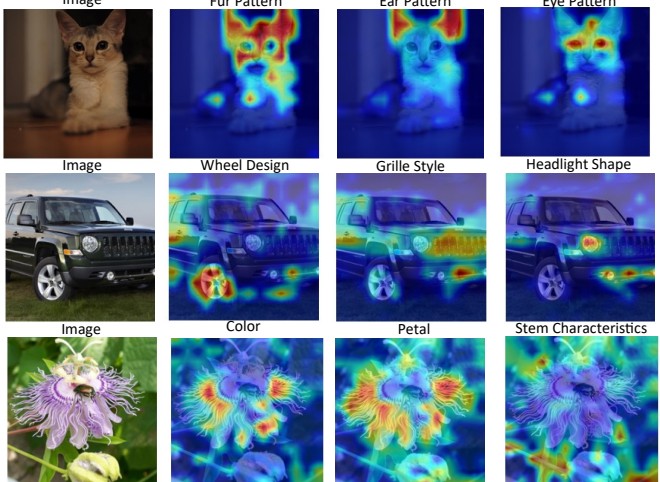

Table 3: Effects of the Tree of Attributes.

| Des. Org. | Unstructured | Ours |
|---|---|---|
| Base | 82.89 | **84.75** |
| Novel | 75.32 | **77.63** |
| HM | 78.93 | **81.04** |

Table 4: Effects of domain experts.

| Align. Token | CLS | Ours |
|---|---|---|
| Base | 83.89 | **84.75** |
| Novel | 76.85 | **77.63** |
| HM | 80.22 | **81.04** |

Figure 3: Visualization of the class activation maps.

Table 5: Effects of the number of experts.

| Attrs. Num. | 1 | 2 | 3 | 4 | 5 | 6 | 7 | 8 | Ours |
|---|---|---|---|---|---|---|---|---|---|
| Base Acc. | 83.20 | 83.97 | 84.1 | 84.41 | 84.45 | 84.62 | 84.66 | 84.74 | **84.75** |
| Novel Acc. | 74.90 | 76.20 | 76.35 | 77.06 | 77.13 | 77.17 | 77.35 | 76.67 | **77.63** |
| HM | 78.83 | 79.90 | 80.04 | 80.57 | 80.63 | 80.72 | 80.84 | 80.50 | **81.04** |

## 4.3 Ablation Study

**Effects of Tree of Attribute.** A core inquiry is whether structuring descriptions into a Tree of Attribute (ToA) offers advantages over an unstructured aggregation of LLM-generated descriptions. To evaluate, we revert to aligning a mixed, unstructured set of descriptions with the CLS token - a common practice in prior studies [25, 19, 40, 52], while keeping the same number of visual prompt tokens. According to Table 3, substituting the ToA with an unstructured set results in significant performance decreases of 1.86%, 2.31%, and 2.11% across the average base, novel, and their harmonic mean performances, respectively. This stark contrast underscores the ToA's critical role in enhancing model efficacy.

**Effects of Learning through Domain Experts.** Further, we examine the impact of substituting the CLS token with visual expert tokens for learning fine-grained attributes, commonly adopted in in previous works [25, 19, 40, 52]. Our findings (Table 4) reveal improvements of 0.89%, 0.78%, and 0.82% in the average base, novel, and harmonic mean accuracies, respectively, upon integrating visual expert tokens. These results support the notion that domain-specific, learnable tokens enhance the model's ability to grasp fine-grained details by focusing on distinct aspects of the image, as opposed to the CLS token's global focus.

**Effects of Number of Attributes.** In our framework, the selection of attributes is dynamically determined by LLMs, leading to variability across different datasets. This adaptability stands in contrast to a static approach where the number of attributes is uniformly set across all datasets. To understand the impact of this variability, we explore how altering the number of attributes from 1 to 8 influences model performance. Our findings, detailed in Table 5, reveal a performance improvement trend as the number of attributes increases, with an optimal peak at 7 attributes before a slight decline at 8. However, crucially, across all fixed-attribute scenarios, none matched the performance achieved through our method's dynamic attribute determination. These results underscore the importance of an adaptive approach to attribute selection, as opposed to a one-size-fits-all strategy.

**Design choice of the vision-conditional pooling layer.** Lastly, we ablate the design of the pooling layer, starting from the naive training-free average pooling, to the attention-based pooling mechanism with condition on the input image. Compared to average pooling, VCP demonstrates a performance gain of 1.08% in the average harmonic mean. Furthermore, when compared with attention-based max pooling, which selects a single description per attribute according to the attention score in Eq. (4),

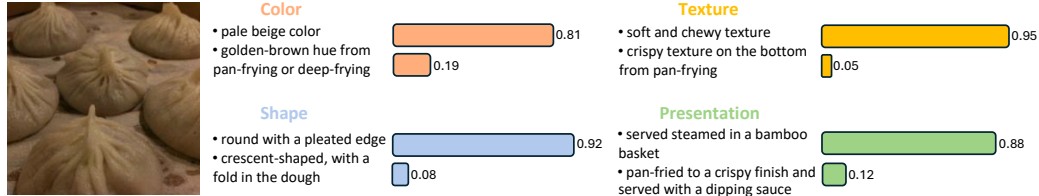

Figure 4: Visualization of the attention weights in the VCP layer for an example "dumplings" image.

Table 6: Design choice of the pooling layer.

| Pooling Method | Base Acc. | Novel Acc. | HM |
|---|---|---|---|
| Attn. Max Pooling | 82.90 | 76.36 | 79.49 |
| Average Pooling | 83.18 | 76.98 | 79.96 |
| VCP (Ours) | **84.75** | **77.63** | **81.04** |

VCP maintains a superior advantage of 1.55% in average harmonic mean. These outcomes attest to the VCP layer's integral role in finetuning attribute relevance to the visual context, substantiating its design and implementation within our model.

## 4.4 Visualization

**Expert tokens focus on attribute-related regions.** We further investigate the effects of vision domain experts by visualizing their class activation maps from three illustrative examples using GradCAM [37], as shown inFig. 3. These visualizations underscore the precision with which each expert token concentrates on the image regions pertinent to its designated attribute. Take the first cat image as an example. The "fur pattern" expert distinctly highlights the animal's fur texture, whereas the "ear" and "eye" experts focus precisely on the respective anatomical features. This pattern of attribute-specific attention is consistent across the evaluated examples, reinforcing the conceptualization of expert tokens as dedicated "domain experts" within the visual field.

**VCP layer pools the most applicable descriptions.** The inherently interpretable nature of the VCP layer, thanks to its attention mechanism, allows for insightful visualizations of its operational process. Through the examination of attention weights assigned by the VCP layer to different attributes in a given image, we elucidate the layer's capability to discern and prioritize the most applicable descriptions. As illustrated in Fig. 4 with a "dumplings" image, the VCP layer adeptly allocates higher attention weights to descriptions accurately reflecting the observed instance (e.g., assigning weights of 0.92 to "round with a pleated edge" under the "Shape" attribute and 0.95 to "soft and chewy texture" under the Texture"). In contrast, less relevant descriptions for the specific image context (e.g., "crescent-shaped" for Shape and "crispy texture from pan-frying" for Texture) receive significantly lower weights. This discernment is crucial, given the class dumplings" encompasses a broad variety of appearances based on cooking methods, yet not all descriptions are fitting for every instance. These visualizations compellingly demonstrate the VCP layer's effectiveness in refining description relevance, thereby enhancing the model's interpretative alignment with the visual content.

## 5 Conclusion

This paper introduces Tree of Attribute Prompt learning (TAP), a novel method that integrates detailed, LLM-generated descriptions within VLMs, achieving state-of-the-art performance in both base-to-novel generalization and few-shot image classification tasks across 11 diverse datasets. TAP leverages a hierarchical "Tree of Attribute" framework, distilling structured knowledge graphs from LLMs for nuanced representation of visual concepts, and employs learnable "domain expert" tokens and a vision-conditional pooling module for optimal image-text alignment. While promising, we note that the reliance on LLMs presents challenges in fine-grained datasets where similar classes require nuanced differentiation, in which cases LLMs generate identical descriptions for distinct classes, impacting novel class prediction performance. It highlights the current limitations of LLMs in discerning highly fine-grained distinctions. Addressing this challenge through enhanced LLM capabilities or alternative strategies will be a key focus of future research.

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

# A  Appendix

## A.1  Model regularization

Denote the frozen image feature from CLIP vision encoder as $\mathbf{f}^v$, the frozen text feature for description $d$ from CLIP text encoder as $\mathbf{f}_d^t$, and the zero-shot logit prediction from CLIP as $\hat{y}$. Additionally, denote the trained image feature as $\tilde{\mathbf{f}}^v$, the trained text feature for description $d$ as $\tilde{\mathbf{f}}_d^t$, and the logit prediction from attribute $a$ after training as $\tilde{y}_a$. The losses are as follows:

$$L_{L_1-V} = ||\mathbf{f}^v - \tilde{\mathbf{f}}^v||_1 \tag{8}$$

$$L_{con-T} = -\sum_{d\in\mathcal{D}} \left( \frac{1}{2}\log\frac{\exp(cos(\mathbf{f}_d^t, \tilde{\mathbf{f}}_d^t))}{\sum_{k\in\mathcal{D}_s}\exp(cos(\mathbf{f}_d^t, \tilde{\mathbf{f}}_k^t))} + \frac{1}{2}\log\frac{\exp(cos(\mathbf{f}_d^t, \tilde{\mathbf{f}}_d^t))}{\sum_{k\in\mathcal{D}_s}\exp(cos(\mathbf{f}_k^t, \tilde{\mathbf{f}}_d^t))} \right) \tag{9}$$

$$L_{KL-attr} = \frac{1}{|\mathcal{A}|}\left( \sum_{a\in\mathcal{A}} \mathcal{D}_{\mathcal{KL}}(\hat{y}, \tilde{y}_a) \right) \tag{10}$$

The regularization loss is then:

$$L_{reg} = \mu_1 L_{L_1-V} + \mu_2 L_{KL-attr} + \mu_3 L_{con-T}, \tag{11}$$

Our overall training objective is thus given by:

$$L_{\text{total}} = L_{\text{class}} + L_{\text{reg}} \tag{12}$$

## A.2  Additional implementation details

We use PyTorch [31] to implement all experiments on a single NVIDIA A100-80GB GPU. Our code is developed based on the implementation of CoOp [54], which is available at https://github.com/KaiyangZhou/CoOp and released under the MIT license. Our code is also released under the MIT license. Baseline results for both base-to-novel generalization and few-shot classification are taken from their respective publications. For the "global context" attribute which is aligned with the CLS token in the vision encoder, we use the following 7 selected templates provided in [33].

`"itap of a {class}."`

`"a bad photo of the {class}."`

`"a origami {class}."`

`"a photo of the large {class}."`

`"a {class} in a video game."`

`"art of the {class}."`

`"a photo of the small {class}."`

## A.3  Prompts for Tree-of-Attribute generation

As introduced in Section 3.3, we generate the Tree-of-Attribute with the following three steps: 1) Attribute Generation, 2) In-Context Example Generation, and 3) Description Generation for All Classes. The prompts for each step are as follows:

**1) Attribute Generation:**

*{Dataset Description.}*

*Visual attributes refer to observable, describable features of the images that can include color, shape, size, texture, and any specific patterns or markings, which can help differentiate between classes for the dataset. They*

*should be consistently observable across multiple images of the same class. Your task is to generate a list of visual attributes (less than 10) for the {Dataset Name} dataset. Ensure this list is clear, concise, and specific to the dataset's needs. Avoid generic attributes that do not contribute to distinguishing between classes.*

**2) In-Context Example Generation**

*Describe describe what a "{Random Class Name}" class in the {Dataset Name} dataset look like using the generated visual attributes.*

*You must follow the following rules:*

*1. For each visual attribute, describe all possible variations as separate sentences. This approach allows for a detailed and clear presentation of each attribute's range.*

*2. Provide a maximum of five descriptions for each visual attribute to maintain focus and relevance. Also, aim to provide at least two descriptions to ensure a comprehensive overview of the attribute.*

*3. The descriptions should provide clear, distinguishable features of each class to support image classification tasks.*

*4. Descriptions for each attribute are independent from each other, and they should not serve as context for each other.*

*5. Each description describes an image independetly. If certain description is possible for a class, please just list that description, and do not use words like "may have" or "sometimes have".*

*6. Reply descriptions only. Do not include any explanation before and after the description.*

*7. The descriptions should follow the format of "classname, which ...", where "..." is the description of the visual attribute.*

**3) Description Generation for All Classes**

*{Dataset Description.}*

*Your task is to write detailed descriptions for various classes within the {Dataset Name} dataset, using the provided visual attributes such as color and shape. These descriptions will help in accurately classifying and understanding the unique features of each class.*

*You must follow the following rules:*

*1. For each visual attribute, describe all possible variations as separate sentences. This approach allows for a detailed and clear presentation of each attribute's range.*

*2. Provide a maximum of five descriptions for each visual attribute to maintain focus and relevance. Also, aim to provide at least two descriptions to ensure a comprehensive overview of the attribute.*

*3. The descriptions should provide clear, distinguishable features of each class to support image classification tasks.*

*4. Descriptions for each attribute are independent from each other, and they should not serve as context for each other.*

*5. Each description describes an image independetly. If certain description is possible for a class, please just list that description, and do not use words like "may have" or "sometimes have".*

*6. Reply descriptions only. Do not include any explanation before and after the description.*

*7. The descriptions should follow the format of "classname, which ...", where "..." is the description of the visual attribute.*

*Q: Describe what a "{Random Class Name}" in the {Dataset Name} look like using the following visual attributes: {Visual Attributes from Step 1.}*

*A: {Answer from Step 2.}*

*Q: Describe what a "{Target Class Name}" in the {Dataset Name} look like using the following visual attributes: {Visual Attributes from Step 1.}*

*A:*

In the prompt templates, *"Dataset Description"* is the description of the dataset from their official website, *"Random Class Name"* is a randomly sampled class name in the dataset for in-context example generation, and *"Target Class Name"* is the class name of interest for the current query. While step 1 and 2 are made in two consecutive calls to provide contexts which are queried once per dataset, step 3 is queried independently for

each of the remaining classes in the dataset. Human review is performed after step 2 to ensure a high-quality set of attributes and in-context example.

## A.4 Potential societal impacts

While our work primarily focuses on advancing prompt learning in vision-language models, it's crucial to acknowledge the potential broader societal implications of such advancements. On the positive side, TAP could lead to more efficient and accurate image understanding systems, benefiting various domains. For instance, it could enhance accessibility for visually impaired individuals by providing more detailed descriptions of visual content. Furthermore, improved visual understanding could contribute to more effective content moderation, mitigating the spread of harmful online materials. However, these advancements also present potential risks. LLMs used for description generation can perpetuate existing societal biases present in their training data, leading to biased outcomes in image recognition. Moreover, sophisticated VLMs could be misused to create misleading visual content, contributing to misinformation and manipulation. The enhanced ability to analyze and understand images also raises privacy concerns, particularly in surveillance contexts where personal information could be extracted from visual data. Addressing these potential negative impacts necessitates careful consideration of bias mitigation techniques during LLM training, promoting transparency and explainability in VLM decision-making, and establishing ethical guidelines for responsible development and deployment of such technologies.

