# OpenReview forum: "Tree of Attributes Prompt Learning for Vision-Language Models"
_NeurIPS.cc/2024/Conference — Submitted to NeurIPS 2024_

### Official Review · Reviewer_Vt6k · 2024-07-06

**Soundness:** 4
**Presentation:** 3
**Contribution:** 3
**Rating:** 6
**Confidence:** 5

**Summary:**

This paper proposes Tree of Attributes Prompt learning (TAP). Unlike previous works that rely on unstructured class descriptions, this approach distillates structured knowledge graphs associated with class names from LLMs. Text/vision prompts and vision-conditional pooling module are designed to extract instance-specific text features. Extensive experimental results demosntrate its improved performances.

**Strengths:**

- Overall, the idea of distillating structured knowledge from LLMs in the task of prompt learning is new and interesting.

- The paper designed an effective prompt learning framework to capture fine-grained attributes, using vision expert tokens and vision-conditional pooling layer.

- The illustrated way to generate structure tree of attribute from LLMs can also be used in other tasks.

- From the experiments, using structured knowledge leads to better performances than unstructured descriptions in base-to-novel and few-shot classification tasks.

- The visualization of class activation maps and attention weights look good. The paper is well written and easy to follow.

**Weaknesses:**

- Apart from the new framework, the method highly relies on the quality of tree of attribute generated with GPT-3.5-turbo. There is no study on the robustness aganist different LLMs, different generation prompts, or varying attribute sets.

- The loss includes a model regularization and its effectiveness is not discussed.

- In Figure 2, it is not too clear to me about $I_1 T_1$, $I_2 T_2$,etc. They seem not be discussed in the text parts.

**Questions:**

- In Table 6, what is the difference between Attn. Max Pooling and VPC? Is that the former selects one most similar while the latter uses a soft weighted sum?

- In Table 5, why using an adaptive number of attributes is better than a fixed 8 number of experts. When increasing the number of experts from 1 to 8, did the authors observe some patterns in the order of added attributes? Say, at the beginning some general attributes, then finer ones, and later irrelevant ones?

- In Ln 189, could the authors clarify 'deep prompting'. Does it mean, in additional to vision prompt tokens at the input layer, there are other vision prompt tokens inside vision encoder not plotted in Figure 2?

- In Eq. 7, could the authors clarify how $\alpha=0.4$ is chosen, and compare the performances of using CLS token only $\alpha=1.0$?

- In Ln 262, how to revert to unstructured set of descriptions? Are they converted from the same tree of attributes by keeping all attributes and descriptions?

**Limitations:**

One limitation is its reliance on LLMs (GPT) to generate the tree of attribute. When generating more complex responses, it is challening to ensure the quality and variances. How to keep a balance between the diversity of attribute sets and relevancy of attributes to classification is important.

---

> ### Author Rebuttal · Authors · 2024-08-07
>
> Thank you for the constructive feedback and the positive assessment of our work! We address the detailed concerns below.
>
> ### W1: LLM Robustness
>
> Thank you for highlighting this concern. We note that since our use of LLM is mainly retrieving simple facts without requiring complex capabilities, the quality of the generated attributes is generally reliable.
>
> 1. Robustness Against Different LLMs
>
> We regenerated the descriptions using a small LLM, Qwen-2-7B-Instruct [1], and obtained comparable results:
>
> | Base | Novel | HM |
> | - | - | - |
> | 84.68 | 77.31 | 80.83 |
>
> 2. Robustness Against Different Generation Prompts
>
> We reran ToA generation pipeline using prompts rewritten by ChatGPT. This process resulted in a slightly different set of attributes. The results were as follows:
>
> | Base | Novel | HM |
> | - | - | - |
> | 84.93 | 77.20 | 80.88 |
>
> These results demonstrate that our method maintains its performance across different LLMs and generation prompts. The consistency in results indicates that our method's effectiveness does not heavily rely on the choice of LLM, given its straightforward nature. As LLM capabilities continue to improve, we anticipate further enhancements in robustness.
>
> ### W2: Effectiveness of Model Regularization
>
> Thank you for pointing this out. We conducted an additional experiment without regularization, while reducing the learning rate to 1/4 of the original values to avoid overfitting. The results are as follows:
>
> | Base | Novel | HM |
> | - | - | - |
> | 83.37 | 75.82 | 79.42 |
>
> As expected, the performance significantly decreases without model regularization, which aligns with findings from previous works [2,3]. For example, MaPLe, which did not use model regularization, reported Base: 82.28, Novel: 75.14, HM: 78.55, whereas PromptSRC, which added model regularization to MaPLe, achieved an average HM of 79.97. These results demonstrate that model regularization is crucial for prompt tuning to prevent overfitting and catastrophic forgetting.
>
> Notably, our work outperforms existing methods both with and without model regularization, showing better performance than MaPLe when regularization is not used, and surpassing PromptSRC when it is used.
>
> ### W3: Clarity of Figure 2
>
> We apologize for any confusion caused by Figure 2. In the figure, each “color” represents an attribute (e.g., orange -> fur pattern). In each attribute:
> - $I_1$ represents the visual expert token $p^v_a$ of attribute $a$.
> - $T_1$, $T_2$, etc. are $v_c^a$, representing the output of the VCP layer for class 1, class 2, etc., in attribute $a$.
> - $I_1T_1$, $I_1T_2$, etc. represent the cosine similarity calculations between the visual expert token and the corresponding pooled textual features of all classes, which are the prediction logits of attribute $a$.
>
> We calculate prediction logits for each attribute and obtain the final prediction via a weighted sum of all prediction logits (refer to Equation 7). We will refine the figure to make this clearer.
>
> ### Q1: Attn. Max Pooling vs. VCP
>
> Both Attn. Max Pooling and VCP are attention-based pooling, but VCP can be viewed as a "soft" version of Attn. Max Pooling. In Attn. Max Pooling, we calculate the attention score between $p^v_a$ and the description set $D_c^a$ in the same way in first three lines of Equation 4. Instead of obtaining $v_c^a$ using the attention score for a weighted sum of $D_c^a$, we perform a max pooling of $D_c^a$ based on the attention score, selecting the description with the highest attention score to the visual expert token. The better performance of VCP than Attn. Max Pooling shows VCP's flexibility in capturing multiple relevant descriptions, whereas Attn. Max Pooling only pools one description.
>
> ### Q2: Adaptive attribute number
>
> We use an adaptive number of attributes because of variations and granularities between datasets. For instance, 4 attributes (Action Pose, Number of People, Background Setting, Objects Present) may suffice for UCF101 but are inadequate for ImageNet, which has greater class variation.
>
> The generated attributes are generally in the same granularity, such as shape, pattern, etc. for ImageNet and fur pattern, eye pattern, etc. for Pets. Once sufficient attributes for class differentiation are available, additional attributes do not provide further benefits and may increase overfitting risks.
>
> ### Q3: Deep prompting
>
> Yes, deep prompting introduces learnable prompt tokens in every transformer layer, as opposed to shallow prompting, which introduces prompt tokens only at the input level. This is a common trick for improving model's performance used in previous works [2,4,5]. To reduce complexity, we applied deep prompting only to the vision encoder. We apologize for not showing this in Figure 2.
>
> ### Q4: Effect of $\alpha$
>
> $\alpha$=0.4 was empirically found to balance global and local information optimally. When $\alpha$=1.0, the performance significantly decreased:
>
> | Base | Novel | HM |
> | - | - | - |
> | 81.54 | 73.85 | 77.51 |
>
> This shows the importance of expert tokens. Additionally, in Table 4, we tried using CLS token instead of expert tokens to align with each attribute feature during training, but also found it significantly worse. These results support the notion that domain-specific expert tokens enhance the model’s ability to grasp fine-grained details by focusing on distinct aspects of the image, as opposed to the CLS token’s global focus.
>
> ### Q5: Unstructured Set of Descriptions
>
> Yes, they were converted by putting all descriptions from the same tree of attributes in a unstructured set of descriptions.
>
> [1] Yang et al. Qwen2 Technical Report. arXiv:2407.10671 (2024)
>
> [2] Khattak et al. Self-regulating Prompts: Foundational Model Adaptation without Forgetting. ICCV (2023)
>
> [3] Yao et al. Visual-Language Prompt Tuning with Knowledge-guided Context Optimization. CVPR (2023)
>
> [4] Khattak et al. MaPLe: Multi-modal Prompt Learning. CVPR (2023)
>
> [5] Jia et al. Visual Prompt Tuning. ECCV (2022)

---

> > ### Comment · Reviewer_Vt6k · 2024-08-10
> >
> > I would like to thank the authors for their detailed responses, which solved my previous questions.

---

> > > ### Author Response · Authors · 2024-08-12
> > > **Thank you**
> > >
> > > Thank you again for your positive assessment and for taking the time to review our work. Your invaluable feedback has been instrumental in improving our paper.

---

### Official Review · Reviewer_grEg · 2024-07-12

**Soundness:** 2
**Presentation:** 3
**Contribution:** 3
**Rating:** 4
**Confidence:** 4

**Summary:**

This paper proposes a new method called "Attribute Prompt Learning Tree (TAP)" to improve the performance of CLIP on zero-shot and few-shot classification tasks. The authors leverage large language models (LLMs) to generate more descriptive text prompts and introduce a hierarchical tree-like structure to systematically generate and integrate these descriptions, ensuring a layered and comprehensive understanding of the visual content. The method also learns specialized "domain expert" prompt tokens that focus on different visual attributes and uses a vision-based pooling module to extract text features for specific instances. Extensive experiments show that TAP outperforms state-of-the-art methods on zero-shot and few-shot classification tasks across multiple datasets

**Strengths:**

1), The idea that utilizing LLM to generate tree-like prompts makes sense. This structured description approach is significantly different from the existing simple text prompt methods and provides an efficient way to improve VLMs.

2), The image-conditional pooling module looks like good for capturing instance-specific features.

3), Experiments and visualization demonstrate the effectiveness of the proposed model.

**Weaknesses:**

1), TAP introduces many textual and visual prompts, which leads to high computing and time costs. This may limit its applications.

2), TAP first generates hierarchical token prompts, while it seems like TAP does not use such a hierarchical structure to integrate the output of the text encoder. It only uses a pooling strategy to update the text encoder output with the visual feature. That is, TAP also does not utilize these relationships in the prompt graph.

3), TAP can be viewed as a multimodal prompt tuning method. What is the main difference between TAP and MAPLE,  ALIGN.

**Questions:**

Please see above

---

> ### Author Rebuttal · Authors · 2024-08-07
>
> Thank you for your valuable comments! We address the detailed concerns below. We hope that our responses will reflect positively on your final decision.
>
> ### W1: Computational and Time Costs
>
> Thank you for highlighting the concern. We would like to clarify how TAP manages these costs effectively compared to previous works.
>
> 1. Efficient Use of Learnable Parameters: While TAP introduces textual and visual prompts, the number of newly introduced learnable parameters is actually less than in previous works such as PromptSRC [1] and MaPLe [2]. Specifically, methods like PromptSRC and MaPLe utilize "deep prompting," introducing "N" learnable prompt tokens in each layer of both the vision and text encoders. In contrast, TAP employs "deep prompting" only for the vision encoder. For the text encoder, we use "shallow prompting," where learnable prompts are introduced only at the input level. This approach significantly reduces the number of additional parameters for the text encoder.
> 2. Inference Efficiency: At inference time, both the vision and text embeddings in TAP can be independently pre-extracted and saved for future retrieval tasks. This means that once the embeddings are generated, they can be reused without recomputing. In contrast, MaPLe [2] utilizes a vision-language (VL) coupling function, which couples the vision and text encoders. This coupling means that the embeddings cannot be cached independently; different images can alter the text feature embedding due to the VL coupling, necessitating re-inference for new images. TAP avoids this issue, making it more efficient for repeated inferences and retrieval tasks.
>
> ### W2: How TAP uses the hierarchical structure
>
> Thank you for your feedback. We would like to clarify how the hierarchical structure in TAP is utilized to enhance the integration of text encoder outputs and the overall model performance.
>
> Our approach leverages the tree structure in two significant ways:
> 1. Description generation in Top-Down: We first generate a set of attributes from the class name, followed by generating descriptions for each attribute. This hierarchical approach ensures that the descriptions are structured and contextually relevant. Unlike previous works that generate an unstructured set of descriptions, our method organizes them into a coherent tree structure, as illustrated in Figure 1 of the paper.
> 2. Utilization of Tree of Attributes in Bottom-Up:
>     - From leaf nodes to attribute-layer, we use the VCP layer to aggregate descriptions in each attribute to form attribute-level features. These features are then aligned with corresponding visual expert tokens, ensuring that each visual expert token focuses on specific, fine-grained attributes.
>     - From attribute-layer to root node class prediction, we aggregate attribute-level features to make class predictions via a weighted sum of the prediction logits (refer to Equation 7). This process allows the model to utilize the structured relationships within the tree, enhancing the alignment and integration of visual and textual data.
>
> By using a top-down approach to generate the tree and a bottom-up approach to utilize it, TAP effectively integrates hierarchical relationships within the prompt graph. This dual usage ensures that the model leverages both the high-level structure and fine-grained details, leading to improved performance and interpretability.
>
> We hope this clarifies the role and utilization of the hierarchical structure in TAP.
>
> ### W3: Difference with MAPLE and ALIGN
>
> Compared with multimodal prompt tuning methods MAPLE and ALIGN, our method significantly differs from them in two key aspects: 1) Main focus. Our method focuses on augmenting category names with prior knowledge from LLMs to better utilize the rich information inherent in the category names. In contrast, these methods rely solely on the original category names and focus on multimodal feature fusion/alignment. 2) Methodology. Our method constructs a tree of attributes to organize LLM-generated descriptions and leverages this structured information through prompting, whereas MAPLE and ALIGN fail to learn such structured information, as they do not associate textual descriptions with category names.
>
> [1] Khattak et al. Self-regulating Prompts: Foundational Model Adaptation without Forgetting. ICCV (2023)
>
> [2] Khattak et al. MaPLe: Multi-modal Prompt Learning. CVPR (2023)

---

> > ### Comment · Reviewer_grEg · 2024-08-12
> >
> > I thank the authors for their detailed response, and I have read other comments. I decide to keep my rating.

---

> > > ### Author Response · Authors · 2024-08-12
> > >
> > > Dear Reviewer,
> > >
> > > Thank you for your response. Could you please let us know if there are any specific concerns that remain? We are more than happy to address any further issues during the discussion period.

---

### Official Review · Reviewer_1xvn · 2024-07-12

**Soundness:** 4
**Presentation:** 3
**Contribution:** 4
**Rating:** 8
**Confidence:** 5

**Summary:**

This paper propose a method that aiming to align the vision modality with not only the category name but also the whole concept subgraph the noun represents in the knowledge graph. This is achieved by adding a bunch of attributes branches attached to this concept. The authors argue that this integration of attribute knowledge will make the alignment more transferrable and thus result in a good performance boost in terms of zero/few shot results.
Basically, this work focusing on the topic of textual prompt enrichment task that is investigated before but implement in a different manner. Additionally, the proposed method use seperate tokens to learn different aspectrs of attributes of given images, working as 'domain expert'.

**Strengths:**

1. Might be the first work trying to align the vision image with structured data. It is quite interesting considering that most text prompts now are less organized and noisy. And structured data, as pointed out in the recent research of LLM, may lead to better reasoning skill for a foundation model.
2. The proposed vision-conditional pooling can help the model filter out descriptions that are not direct appeared in the image.
3. Recieve good results on different classification datasets with the model trained with this method.

**Weaknesses:**

1. The attributes description is generated by the LLM, which could contain hallucinated content. While there are many reliable sources of knowledge such as wikipedia or conceptNet, this paper seems skip these sources to obtain some accurate attributes.
2. Though this paper decide to use a tree structure to represent the concept. The built tree is not encoded in a structure-awared manner. They are still feed as langauge tokens to the LLMs.
3.  in equation (5), what is $v_y^a$ stands for?
4. The author argued that the vision-conditional pooling, which is bascially a cross attention layer between the visual and language modal. The authors believe this this design will make the model filter out non-exisiting material in the text description. However, we know that due to the quirk of softmax function. You can never make some tokens attention to be '0'. Thus, the model is learning some spurious correlation aftertall.

**Questions:**

1. Why structured description is so important in your presumption? Given that only the text prompts are structured but visual data are not, will this fact hinder the model to learn a structured in-detail alignment?
2. How do you make sure one expert token will only learn from one attribute?
3. Can the model trained this way also work well on the downstream tasks?

**Limitations:**

Not applicable.

---

> ### Author Rebuttal · Authors · 2024-08-07
>
> Thank you for the constructive feedback and the positive assessment of our work! We address the detailed concerns below.
>
> ### W1: Potential for Hallucinated Content from LLM
>
> Thank you for your insightful comment. We considered using Wikipedia for description generation; however, this approach still necessitates an LLM to extract structured descriptions from extensive Wikipedia pages, which may still result in hallucinated content. Additionally, processing long Wikipedia pages can be resource-intensive. Since our task is essentially retrieving simple facts from LLM that does not require sophisticated reasoning or planing capabilities, the risk of hallucination is significantly reduced.
>
> ### W2: Structure-Aware Encoding of the Tree
>
> We apologize for any confusion caused. Our tree structure serves dual purposes:
>
> 1. Generation of Descriptions: The tree structure guides the generation of descriptions by first generating a set of attributes from the class name and then generating descriptions for each attribute. This approach contrasts with previous works that generate an unstructured set of descriptions (refer to Figure 1).
> 2. Utilization of the Tree of Attributes: The generated Tree of Attributes is used as follows:
> The VCP layer aggregates descriptions in each attribute (leaf nodes) to form attribute-level features, which align with the visual expert tokens.
> These attribute-level features are then aggregated to make class predictions via a weighted sum of the prediction logits (refer to Equation 7).
>
> Thus, our approach generates the tree in a top-down manner and utilizes it in a bottom-up manner.
>
> ### W3: Equation (5) clarification
>
> We apologize for the confusion. $y$ in $v^a_y$ represents the ground truth, thus $v^a_y$ represents attribute $a$'s VCP-pooled text embedding of the ground truth class.
>
> ### W4: Potential for Spurious Correlation in Vision-Conditional Pooling
>
> We acknowledge your concern. We note this issue to be a common problem of the attention mechanism. To address this, we experimented with "Quiet Attention" proposed by Evan Miller [1], obtaining comparable results:
>
> |Base|Novel|HM|
> |-|-|-|
> |84.65|77.48|80.90|
>
> This demonstrates that our current implementation of the VCP module, which emphasizes relevant descriptions while de-emphasizing irrelevant ones (as showcased in Figure 4), is effective for the current study.
>
> ### Q1: Structed Description
>
> We argue that visual data implicitly has a tree structure. The raw image can be seen as the root node, the expert tokens as the attribute layer, and the text description features linked to the expert tokens by the VCP module as the leaf nodes. In this framework, the image corresponds to the class label in text, visual expert tokens align with text attributes, and leaf nodes are shared between vision and text via the VCP layer. This tree structure facilitates fine-grained alignment between visual and textual data.
>
> ### Q2: One Expert for One  Attribute
> We make sure one expert token only learn from one attribute by only aligning one expert token to one specific attribute. That is, we have the same number of expert tokens as the number of attributes in the generated Tree of Attributes, and each expert token aligns with one of the attribute via contrastive learning.
>
> ### Q3: Downstream tasks
> Yes, the model works well on downstream tasks. Our base-to-novel experiments, where the model is trained on base classes and tested in zero-shot on novel classes, demonstrate superior performance compared to other baselines. Additionally, we conducted a cross-dataset experiment, where we trained on ImageNet with 16 shots and tested in zero-shot on the remaining 10 datasets. As shown in Table 1 of the Global response, TAP outperforms PromptSRC by 1.03% on ImageNet and 0.75% on average across the other 10 datasets. This indicates that our model generalizes well to downstream tasks.
>
> [1] Evan Miller. Attention Is Off By One.  (2023)

---

> > ### Comment · Reviewer_1xvn · 2024-08-12
> >
> > 1. Many works have validate that including knowledge sources or using RAG can alleviate hallucination problem. So I believe if you can insert a paragraph from a reliable sources to the LLM as context, the LLMs will generate more reliable results comparing to direct doing QA.
> > 2. Thank you for your explanation. From my understanding, you grouped a collection of discription for each attribute by using VCP and each VCP is aligned with an Expert token to reflect the two-level hirerachical structure, is that correct? I was expecting there is a Graph embedding or something close to embed the tree structure. But I still find this implementation is interesting. Good job.
> > 3. I'd like to learn more about the visiual Expert tokens in 3.4. So, you said that you have A independent tokens serve as Expert tokens. And you insert these vision expert tokens before the image patch sequence like what VPT. Then I wonder, are you inserting all the expert tokens together into the image embedding sequences? If that is the case, when doing the cross attention mentioned in 3.5, how do you make sure only the relevant Expert tokens is doing cross-pooling with the relevant attributes, as described in equation 4. Please answer this question clearly and in detail. I will consider to raise my rating if I received a satisfying response.

---

> > > ### Author Response · Authors · 2024-08-12
> > >
> > > 1. Thank you for your suggestion. We agree that incorporating reliable sources as context before querying the LLM could generate more accurate and reliable results. We will certainly explore this direction in our future work to further enhance the robustness of our method.
> > >
> > > 2. Thank you for your compliment and understanding. You are correct that our approach groups a collection of descriptions for each attribute using VCP, and each VCP is aligned with an expert token to reflect a hierarchical structure. To clarify further, the structure in our method is actually three levels:
> > >
> > > - Class Name (Root Node) → Attributes (Intermediate Layer) → Descriptions (Leaf Nodes)
> > >
> > > VCP helps aggregate from descriptions to attributes (from leaf nodes to the attribute layer). Then, the prediction fusion via weighted sum aggregates the attribute predictions into the final class prediction (from the attribute layer to the root node). While we did not use an explicit graph embedding to represent the tree structure, we implemented the hierarchical structure in a more implicit manner through this approach. We appreciate your positive feedback on this implementation.
> > >
> > > 3. Thank you for providing me the opportunity to clarify. Yes, all expert tokens are indeed inserted together into the image embedding sequences, similar to what VPT did. To ensure only the relevant expert token is doing cross-attention, we used a separate VCP module for each expert token.
> > >
> > > Concretely,
> > > - The number of attributes in the Tree of Attributes (ToA) equals the number of expert tokens added, which also equals the number of VCP modules.
> > > - Each attribute description set is aggregated using a dedicated VCP module that operates solely for that attribute. The pooled embedding from this VCP module is then aligned with the specific expert token associated with that attribute.
> > > - In the VCP module (batch size is omitted for simplicity in the following notation):
> > >     - The query is the expert token of shape $1\times D$ (where $D$ is the embedding dimension).
> > >     - The key is the set of descriptions in this attribute, with shape $N\times D$ (where $N$ is the number of descriptions).
> > >     - The resulting pooled embedding, after cross-attention, is of shape $1\times D$, which aligns directly with the expert token (also of shape $1\times D$).
> > >
> > > This design ensures that each expert token only interacts with its relevant attribute.
> > >
> > > I hope this detailed explanation clarifies your concerns. Please feel free to ask any further questions, and I sincerely appreciate your consideration in potentially raising the rating.

---

> > > > ### Comment · Reviewer_1xvn · 2024-08-12
> > > >
> > > > Thank you for your response. For 3, I'd like to ask further questions as following:
> > > > 1. I understood that you are using a expert token to do cross-attention for only one group of description. But my problem is, during implementation, how do you select a SPECIFIC expert token to do attention? In your response, you said you insert a bunch of expert tokens into the image embedding sequences. From my understanding, the sequence of are with the shape $E \times D$ now, where E is the size of expert tokens. When you do cross-attention, all the expert tokens will have exposure to the description embedding, isn't it? How do you avoid such exposure? In other words, how to you implement the one expert to one attribute part in your codes? Would you like to share that part?
> > > > 2. Another question is, you mentioned in your paper said that VCP module is also responsible for filtering out non-related descriptions. Do you have any experiments to show that VCP can actually filtering out those descriptions? Because we all know that some not all descriptions will present in a same image for an object.

---

> > > > > ### Author Response · Authors · 2024-08-12
> > > > >
> > > > > Thank you for your follow-up question.
> > > > >
> > > > > 1. I apologize for any confusion caused. In our implementation, each expert token is aligned with only one specific attribute's descriptions, ensuring no cross-interference with other attributes. Here’s how this works in practice:
> > > > >
> > > > > - Text Input as a Dictionary: The text input, representing the Tree of Attributes (ToA), is formatted as a dictionary: ToA = {'color': [color description set], 'shape': [shape description set], ...} After processing through the text encoder, the ToA is embedded as a tensor of shape $E\times N\times D$, where E is the number of attributes (e.g., color, shape), N is the number of descriptions per attribute (to ensure equal length, we apply zero-padding).
> > > > > - Expert Tokens Interaction: Each expert token from the vision encoder is aligned with its corresponding attribute's descriptions. This is achieved by iterating over each attribute and ensuring that the expert token interacts only with the relevant attribute's descriptions. Below is a pseudocode illustration of this process:
> > > > > ```
> > > > > text_embed = text_encoder(ToA)  # ToA is a dictionary, text_embed is E x N x D
> > > > > image_tokens = vision_encoder(image)  # E x D
> > > > > pooled_embeds = []
> > > > > for i in range(E):
> > > > >     expert_token = image_tokens[i]  # 1 x D
> > > > >     attribute_embed = text_embed[i]  # N x D
> > > > >     pooled_embed = VCP(expert_token, attribute_embed)  # 1 x D
> > > > >     pooled_embeds.append(pooled_embed)
> > > > > ```
> > > > >
> > > > > This process ensures that each expert token is aligned and performs cross-attention only with its corresponding attribute descriptions. The attribute of a vision expert token is determined by the corresponding group of textual descriptions. Although we omitted the handling of multiple classes and batch size for simplicity, in our actual implementation, we used tools like einops to make the process more efficient. We will release our code.
> > > > >
> > > > > 2. Yes, we have conducted both quantitative and qualitative experiments to demonstrate that the VCP module can effectively filter out irrelevant descriptions.
> > > > >
> > > > > Quantitatively, in Table 6, we ablated the design choices for the pooling layer, comparing VCP against average pooling and attention-based max pooling which only pools one description that has the highest attention score per attribute. The results show that VCP significantly outperforms these alternatives.
> > > > >
> > > > > Qualitatively, we visualized the attention weights of VCP in Figure 4. We observe that the model assigns higher attention to relevant descriptions while giving near-zero attention to incorrect or unrelated ones. This behavior illustrates VCP's ability to focus on the descriptions that are most pertinent to the image content, effectively filtering out those that do not apply.
> > > > >
> > > > > I hope this explanation clarifies the implementation. If you have any further questions or need more details, I’m happy to provide additional information. Thank you for your consideration, and I look forward to your feedback.

---

> > > > > > ### Comment · Reviewer_1xvn · 2024-08-13
> > > > > >
> > > > > > 1. Great! Thank you for your detailed explaination and patience. Now I fully comprehend this design, and find it quite inspiring. Just one more concern, an iterative training approach would be less efficient. Is it possible to improve this by using some sort of mask techniques?
> > > > > > 2. Sorry for I didn't notice that Figure and results. Now I think you address all my concerns perfectly. Therefore, I will increase my rating accordingly. Good luck.

---

> > > > > > > ### Author Response · Authors · 2024-08-13
> > > > > > > **Thank you**
> > > > > > >
> > > > > > > Thank you for your recognition of our work and for increasing your rating.
> > > > > > >
> > > > > > > Regarding your question about the efficiency of the iterative training approach:
> > > > > > >
> > > > > > > To improve efficiency, we used einops rearrange to manage multiple classes and batch sizes, along with zero-padding and attention masks to handle varying numbers of descriptions across attributes. We note that these implementations can already maintain a high training effiency. Thank you for your nice suggestion. We will certainly further explore more strategies to improve training efficiency.
> > > > > > >
> > > > > > > Thank you again for your valuable feedback and for the increase in your rating.

---

### Official Review · Reviewer_ZE3g · 2024-07-17

**Soundness:** 2
**Presentation:** 3
**Contribution:** 2
**Rating:** 4
**Confidence:** 4

**Summary:**

The TAP method structures textual descriptions in a hierarchical “concept-attribute-description” format, effectively creating a knowledge graph from large language models (LLMs) for each category name. This structure allows for a more comprehensive and detailed understanding of the visual content. The paper reimagines learnable prompt tokens as "domain experts," each specializing in different aspects of the image, supplemented by a global perspective provided by the CLS token. To address potential misalignment between general descriptions and specific image content, the paper introduces a vision-conditional pooling module. This module extracts instance-specific text features, ensuring optimal image-text alignment.

**Strengths:**

The proposed method incorporates structured tree of attribute into prompt tuning that provide richer supervisory information compared to unstructured attribute information. A set of experiments has been conducted, and the results look promising.

**Weaknesses:**

One major limitation of the method is that it requires human review to "ensure the quality of the example" (L175). Recall that one major advantage of prompt tuning is that it can adapt large models quickly to specific tasks. However, the requirement of human reviewing in the proposed method is not consistent with this goal. In addition, it is not clear how many human efforts are needed here, and how to handle the potential human bias in quality evaluation.

The paper lacks cross-dataset experiments, which is typically provided in existing PT papers. The results are important to examine the domain generalization capability of the method.

For training details, different learning rates were used for different datasets, however, existing methods typically use a same LR for all datasets. From this point, the comparison is somewhat unfair.

**Questions:**

In Section 3.3, for attribute generation, what type of dataset information is given to the large model?

In Figure 4, each image is accompanied by only two descriptions. Are all images described using two sentences each?

In this paper, it mentions that the method can capture subtle differences between attributes. Could you provide a relevant example?

**Limitations:**

yes

---

> ### Author Rebuttal · Authors · 2024-08-07
>
> Thank you for your valuable comments! We address the detailed concerns below.  We hope that our responses will reflect positively on your final decision.
>
> ### W1: Human Review Requirement
>
> We appreciate the reviewer highlighting the concern regarding the human review process in our method. We apologize for any confusion caused by our explanation. The human efforts mentioned refer to the process of curating a 1-shot example for in-context learning when prompting LLMs. Unlike previous works [1] that manually curate descriptions, we have streamlined the process by making it semi-automatic. LLMs generate the examples, followed by a brief human review. Typically, the generated examples are sufficiently accurate and require less than 30 seconds per dataset for a quick read-through, as they mostly retrieve simple facts from LLMs. This minimal human involvement ensures quality without significant effort or bias.
>
> ### W2: Cross-Dataset Experiments
>
> We have conducted additional "cross-dataset" experiments where we trained the model on ImageNet with 16 shots and tested it directly on the remaining 10 datasets in a zero-shot manner. The results are presented in Table 1 of the Global response. Compared to PromptSRC, TAP achieved a +1.03% improvement on ImageNet and a +0.75% average improvement across the 10 datasets, demonstrating robust domain generalization capabilities.
>
> ### W3: Use of Different Learning Rates
>
> We understand the concern regarding the use of different learning rates. The primary reason for splitting the datasets into two groups with different learning rates is due to the variability in ease of learning from LLM-generated descriptions.
>
> However, we also note that using different hyperparameters for different datasets is not uncommon in existing prompt-learning papers. For instance, TaskRes [2] used different learning rates for ImageNet and other datasets. Similarly, CoOP [3] and DMN [4] utilized different numbers of epochs for various datasets, and PromptSRC applied different GPA hyperparameters across dataset groups.
>
> ### Q1: Dataset Information
>
> As stated in Appendix A.3, the dataset information provided to the LLM is the description from the dataset's official website. For example, the dataset information for EuroSAT is: "EuroSAT dataset is based on Sentinel-2 satellite images covering 13 spectral bands and consisting of 10 classes."
>
> ### Q2: Number of Descriptions
> We apologize for the confusion. As indicated in our prompts in Appendix A.3, the number of descriptions per class ranges from 2 to 5. The figure is an illustrative example and does not limit the descriptions to two sentences per image.
>
> ### Q3: Relevant Example
>
> At higher hierarchy, the model captures details from different attributes in the image through the alignment of vision expert tokens and corresponding text attribute features, as showcased in Figure 3. At the attribute level, the model captures the variations within the class via the use of the VCP module as showcased in Figure 4.
>
>
> [1] Menon et al. Visual Classification via Description from Large Language Models. ICLR (2023)
>
> [2] Yu et al. Task Residual for Tuning Vision-Language Models. CVPR (2023)
>
> [3] Zhou et al. Learning to Prompt for Vision-Language Models. IJCV (2022)
>
> [4] Zhang et al. Dual memory networks: A versatile adaptation approach for vision-language models. CVPR (2024)
>
> [5] Khattak et al. Self-regulating Prompts: Foundational Model Adaptation without Forgetting. ICCV (2023)

---

> > ### Comment · Area_Chair_if7Y · 2024-08-13
> > **Please respond to the authors' rebuttal**
> >
> > Dear Reviewer ZE3g,
> >
> > Thank you again for reviewing this paper. Since the reviewer-author discussion phase is closing soon, could you please respond to the authors' comments?
> >
> > Best,
> >
> > AC

---

> > ### Comment · Reviewer_ZE3g · 2024-08-13
> > **Reply to rebuttal**
> >
> > Thanks for the rebuttal. It addresses some of my concerns but one major concern still exists. One limitation of the method is that it requires many tunings specific to each dataset in order to achieve good performance. The tuning process involves two key aspects:
> >
> > __Human Intervention (W1)__:  thought the required human effort is minimal, its impact on model performance remains unclear after the rebuttal.
> >
> > __Learning Rate Variability (W3)__: The method employs different learning rates for different datasets. While TaskRes is trained in a similar manner, the experiments lack comparative analysis with this approach. Additionally, further discussion is needed on how to determine the most appropriate learning rate for a given dataset.
> >
> > I expect authors provide additional insights regarding this issue.

---

> > > ### Author Response · Authors · 2024-08-13
> > >
> > > Thank you for your follow-up questions.
> > >
> > > W1. We apologize for any confusion.
> > >
> > > To clarify, the human review stage is designed to ensure the quality of LLM-generated descriptions. In practice, we found the LLM-generated descriptions good enough and no manual editing was involved in this stage. Therefore, even if we remove this stage, our model would still achieve the same results. Additionally, in the LLM robustness experiment requested by Reviewer Vt6k, we regenerated the descriptions using Qwen2-7B-Instruct without any human review due to the limited time during the rebuttal process. The results are as follows:
> > >
> > > | Base | Novel | HM |
> > > |------|-------|----|
> > > | 84.68 | 77.31 | 80.83 |
> > >
> > > These robust results show that the method's performance is maintained.
> > >
> > > W3. We apologize for not including TaskRes as one of our baselines. We compare TAP and TaskRes in the 16-shot setting as follows:
> > >
> > > | Method | ImageNet | SUN | Aircraft | EuroSAT | Cars | Food | Pets | Flowers | Caltech | DTD | UCF | Average |
> > > |--------|----------|-----|----------|---------|------|------|------|---------|---------|-----|-----|---------|
> > > | TaskRes | 73.0 | 76.1 | 44.9 | 82.7 | 83.5 | 86.9 | 92.4 | 97.5 | 95.8 | 71.5 | 84.0 | 80.8 |
> > > | TAP | 73.8 | 77.3 | 50.4 | 91.9 | 85.4 | 87.5 | 93.9 | 98.1 | 96.7 | 74.9 | 87.2 | 83.4 |
> > >
> > > TAP outperforms TaskRes on all datasets, with an average improvement of 2.6% across the 11 datasets.
> > >
> > > Regarding the determination of learning rates, we apologize for not being clear enough in our rebuttal. We grouped the datasets based on the number of attributes and adjusted the learning rates for vision and text encoders separately based on our intuition to balance generalizability and performance.
> > >
> > > Concretely, for the vision encoder, datasets that have fewer attributes also have fewer learnable expert tokens (thus fewer parameters and lower learning difficulty), we used a larger learning rate (0.006 vs. 0.004) to facilities the learning process.
> > >
> > > For text encoder, the number of learnable text prompts is fixed, where fewer attributes provide fewer text descriptions/data, and thus a smaller learning rate (0.002 vs. 0.004) was used to avoid overfitting.
> > >
> > > We hope this additional context clarifies our approach. Thank you again for your thoughtful feedback and for considering these points in your evaluation.

---

### Official Review · Reviewer_bYva · 2024-07-22

**Soundness:** 3
**Presentation:** 2
**Contribution:** 2
**Rating:** 5
**Confidence:** 4

**Summary:**

This paper proposes a new prompt tuning method for adapting the vision-language model.  The authors design the tree of attribute prompt learning to substitute the categorical description for adapting the vision-language model. A vision-conditional pooling module is proposed to extract instance-specific text features. Extensive experimental results demonstrate the effectiveness of the proposed method.

**Strengths:**

1. A tree of attribute prompt learning method is proposed to guide the adatpation of VLM with the hierarchical semantic information.

2. This paper is well-written and easy to follow.

**Weaknesses:**

1. According to the experiment, the performance improvement of TAP is marginal, e.g., the few-shot performance on most of datasets.  Although the visualization results of VCP layer are impressive, the improvement of this module is also very slight compared to average pooling.

2. The core motivation of this method is learning fine-grained attributes to adapt VLMs. However, similar ideas have been explored in previous works , e.g., APPL[1], MAP[2].  Please discuss the differences.

[1] AAPL: Adding Attributes to Prompt Learning for Vision-Language Models

[2] Multi-modal Attribute Prompting for Vision-Language Models

3. The construction of ToA depends heavily on the prior information on the category of attributes suitable for the dataset. However, one of the most capability of VLM is its zero-shot ability in the open-vocabulary context. What's the performance of the proposed method in the domain generalization setting?


4. The model details in Figure 2 are not presented very clear, especially the input & output streams. This figure should be refined for better clarity.


5. The mechanism behind Equation (5) and the function of VCP needs more clarification. Why conduct constrastive learning between expert token P_a^v and attribute embedding v_c^a generated from P_a^v itself, instead of P_a^v and the embedding of attribute descriptions D?

**Questions:**

Please refer to Weaknesses.

**Limitations:**

Yes.

---

> ### Author Rebuttal · Authors · 2024-08-07
>
> Thank you for your valuable comments! We address the detailed concerns below.
>
>
> ### W1: Marginal Performance Improvement
>
> We appreciate the reviewer's observation.
>
> **a. Generalizability in Base-to-Novel Experiments:**
> The base-to-novel experiment is crucial for evaluating the generalizability of prompt learning methods. According to CoCoOP [1], models can overfit to base classes in a few-shot setting. Our results show that TAP excels in this aspect, indicating robust generalization capabilities.
>
> **b. Significance of Performance Increase:**
> While the few-shot setting is less important, an average performance increase of 0.5% in the few-shot classification is still noteworthy. For context, the improvement from CoOP to ProGrad is 0.8%, and CoCoOP performs nearly 5% **worse** than CoOP in few-shot settings. Thus, the improvements brought by TAP, though appearing small, are significant enough within the domain.
>
> **c. Improvement of VCP Module:**
> Regarding the VCP module, Table 6 indicates that VCP achieves a 1.08% improvement over average pooling. This enhancement is substantial, demonstrating the effectiveness of VCP in refining attribute relevance to the visual context, which translates to better overall performance.
>
> ### W2: Differences between TAP and previous works
>
> The key differences between TAP and existing methods are the structured approach to integrating LLM-generated descriptions and the use of visual expert tokens for fine-grained alignment. As for the two referenced papers, although some similarities exist, significant differences between our work and theirs are evident. Additionally, TAP outperforms them by large margins (TAP HM 81.04 vs. AAPL 76.01 and MAP 79.36).
>
> Regarding AAPL: While both AAPL and TAP aim to enhance vision-language models through attribute integration, they diverge fundamentally in their methodologies and objectives. AAPL focuses on managing learnable prompts via data augmentation, using adversarial token embedding to mitigate bias and enhance attribute-specific learning through augmented image features. However, this approach is constrained by the limitations of image data augmentation, which cannot cover all possible variations. In contrast, TAP leverages LLM-generated knowledge to construct a structured Tree of Attributes. This method explicitly captures fine-grained, instance-specific attributes, such as differentiating between a pan-fried crescent-shaped dumpling and a round steamed dumpling, which image augmentations in AAPL cannot achieve. Thus, TAP's utilization of LLM-generated descriptions allows for a more comprehensive and accurate adaptation of VLMs to diverse and nuanced visual concepts.
>
> Regarding MAP: While both TAP and MAP leverage LLM-generated descriptions and introduce learnable prompts in the vision encoder, several key differences distinguish TAP. First, MAP generates descriptions in an unstructured manner, as depicted in Figure 1(b) of our paper, whereas TAP organizes this information into a structured Tree of Attributes (ToA). Second, although both methods aim to enhance fine-grained alignment, MAP aligns at the individual description level, which can lead to redundancy and misalignment due to similar descriptions falling under the same attribute category or containing irrelevant information for the specific image. TAP, on the other hand, aligns visual expert tokens at a higher "attribute" level, ensuring each token focuses on a specific attribute class. This structured approach mitigates the risk of misalignment and enhances the specificity and relevance of the visual prompts. Furthermore, TAP employs a vision-conditional pooling module to filter out irrelevant descriptions, which is not addressed in MAP, providing a more robust and contextually accurate alignment.
>
> ### W3: Dependency on Prior Information for ToA Construction
>
> We understand the reviewer's concern. To address this, we performed additional "cross-dataset" experiments to evaluate the zero-shot generalization capabilities of our method.
>
> In the experiment, following the common practice, we trained our model on ImageNet under 16-shot and tested it directly on the remaining 10 datasets in a zero-shot manner. The results are presented in Table 1 of the Global response. Compared to PromptSRC, TAP achieved a +1.03% improvement on ImageNet and a +0.75% average improvement across the 10 datasets. These results underscore TAP's robust performance in domain generalization settings, reinforcing its utility in open-vocabulary contexts.
>
> ### W4: Clarity of Figure 2
>
> Thank you for your constructive feedback. In Figure 2, the input for the vision encoder is the tokenized image tokens and learnable visual expert tokens. The input for the text encoder is the generated descriptions and learnable text prompts. Each attribute has its prediction and the fusion of all attributes is treated as the final output. We will update the figure to ensure it clearly illustrates the input and output streams of our model in the revised version.
>
> ### W5: Clarification of equation (5)
>
> We apologize for the confusion caused. In equation (5), $v_c^a$ is the pooled textual embedding from attribute $a$ of class $c$, where pooling is done by the VCP module (cross attention between $p^v_a$ and the descriptions in the attribute description set $D_c^a$). The reason why we didn't contrast $p^v_a$ against $D$ is because $D$ is a set of textual embeddings. Note that the vision-conditional pooling module pools suitable textual descriptions with the visual instance condition so the $v_c^a$, as an instance-specific text feature, can be better aligned with the vision embeddings.
>
> [1] Zhou et al., Conditional Prompt Learning for Vision-Language Models. CVPR (2022)

---

> > ### Comment · Reviewer_bYva · 2024-08-11
> > **Reply to rebuttal**
> >
> > The responses have addressed all my questions.

---

> > > ### Author Response · Authors · 2024-08-12
> > > **Thank you**
> > >
> > > Thank you again for your positive assessment and for taking the time to review our work. Your invaluable feedback has been instrumental in improving our paper.

---

### Author Rebuttal · Authors · 2024-08-07

We sincerely thank all reviewers for your valuable comments. We first reply to questions raised by multiple reviewers and then other questions from every reviewer.

Q1. Model's generalizability. (Reviewer bYva, ZE3g, 1xvn)

To evaluate the generalizability of our model, we conducted an additional cross-dataset experiment where we trained on ImageNet with 16-shot and tested on the remaining 10 datasets in a zero-shot manner. As shown in Table 1, TAP achieved state-of-the-art performance, outperforming PromptSRC by 1.03% on ImageNet and 0.75% on average across the 10 datasets. Notably, TAP is 4.7% better on DTD, 1.2% better on SUN397, and 0.69% better on FGVC Aircraft compared to PromptSRC, demonstrating TAP's superior generalizability.

Table 1. Cross-dataset benchmark evaluation. Trained on the source dataset (ImageNet) with 16-shot and tested on all other target datasets in zero-shot. "Average" represents the average performance across all target datasets.
||ImageNet | Caltech101 | OxfordPets | StanfordCars | Flowers102 | Food101 | Aircraft | SUN397 | DTD | EuroSAT | UCF101 | *Average* |
| - | - | - | - | - | - | - | - | - | - | - | - |  - |
|CoOp|71.51|93.70 |89.14 |64.51 |68.71 |85.30| 18.47| 64.15 |41.92 |**46.39**| 66.55| 63.88|
|Co-CoOp|71.02|**94.43**|90.14 |65.32 |**71.88**|86.06| 22.94 |67.36 |45.73 |45.37 |68.21|65.74|
|PromptSRC | 71.27 | 93.60 | 90.25 | **65.70**| 70.25 | **86.15** | 23.90 | 67.10 | 46.87 | 45.50 | 68.75 | 65.81 |
|TAP|**72.30**|94.30|**90.70**|65.60|70.93|86.10|**24.57**|**68.30**|**50.20**|46.00|**68.90**|**66.56**|

---

> ### Comment · Area_Chair_if7Y · 2024-08-11
> **Reminder -- please reply to rebuttal**
>
> Dear Reviewers,
>
> Thank you for your comments. Please read through all the reviews and the rebuttal and see if authors' responses have addressed your and others' concerns.
>
> Best,
>
> AC

---

### Decision · Program_Chairs · 2024-09-25

**Decision:**

Reject

**Comment:**

This paper proposes a prompt learning method for adapting vision language models, which instructs LLMs to generate a tree of attributes for each category name and learn the hierarchy with vision and text prompt tokens. The tree structure has a hierarchical “concept-attribute-description” format and the prompt tokens are designed to capture the corresponding visual attributes. A vision-conditional pooling module is introduced to extract instance-specific text features. The proposed method is evaluated on 11 datasets.

This paper received mixed reviews. Reviewers identified the merits of this paper, including: 1. the idea is interesting; 2. the paper is well written and easy to follow; 3. the experimental results are promising. They also raised a number of concerns, including: the proposed method requires many tunings specific to each dataset such as learning rate; lack of experimental evaluation in aspects such as domain generalization setting, marginal improvement, and computational cost; some technical designs need to be clarified (e.g., how to control the quality of the generated attribute description and use the tree prompt in a structure-aware manner) and some technical details are not clear; lack of discussions on the differences with related methods.

Although the rebuttal alleviated part of these issues, some serious concerns remained unaddressed. Especially, the experimental comparison can be unfair since the proposed method uses different learning rates for different datasets and the proposed method requires human reviewing. In the post-rebuttal discussion, the authors argued that there is another method (not compared in the original paper) that also uses different learning rates for different datasets, which may not be a valid reason and is still unfair for other compared methods. The authors claimed that the learning rates are determined "based on our intuition", which is not convincing enough. Although the authors showed that the performance of the proposed method can be maintained in some cases even without manual editing, more thorough and fair comparisons are needed. Moreover, it is difficult to include all the necessary discussions and additional results within the small edits allowed for camera-ready. Overall, although the AC agreed that the idea of this paper is interesting, this paper in its current version may not be ready for publication in NeurIPS. The reviewers have provided very detailed feedback and the AC suggests the authors follow their comments closely and submit the revised version to a future ML venue.